# Earth Observation and GIS-Based Analysis for Landslide Susceptibility and Risk Assessment

**Emmanouil Psomiadis** [1,*] , **Nikos Charizopoulos** [1] , **Nikolaos Efthimiou** [2] ,
**Konstantinos X. Soulis** [1] **and Ioannis Charalampopoulos** [3]

[1]  Department of Natural Resources Management and Agricultural Engineering,
   Agricultural University of Athens, 75 Iera Odos st., 11855 Athens, Greece; nchariz@aua.gr (N.C.);
   soco@aua.gr (K.X.S.)

[2]  Faculty of Environmental Sciences, Czech University of Life Sciences Prague, Kamýcká 129,
   165 00 Prague–Suchdol, Czech Republic; efthimiou@fzp.czu.cz

[3]  Department of Crop Science, Agricultural University of Athens, 75 Iera Odos st., 11855 Athens, Greece;
   icharalamp@aua.gr

*  Correspondence: mpsomiadis@aua.gr; Tel.: +30-210-529-4156

**Abstract:** Landslides can cause severe problems to the social and economic well-being. In order to effectively mitigate landslide hazards, the development of detailed susceptibility maps is required, towards implementing targeted risk management plans. This study aims to create detailed landslide susceptibility (LS) and landslide risk (LR) maps of the Sperchios River basin by applying an expert semi-quantitative approach that integrates the Geographic Information Systems (GIS)-based multicriteria analysis and Earth Observation (EO) data. Adopting the analytic hierarchy process (AHP) for a weighted linear combination (WLC) approach, eleven evaluation parameters were selected. The results were validated using a historic landslide database, enriched with new landslide locations mapped by satellite and aerial imagery interpretation and field surveys. Moreover, the landslide risk map of the area was also developed, based on the LS delineation, considering additionally the anthropogenic exposure and overall vulnerability of the area. The results showed that the most susceptible areas are located at the west and south-west regions of the basin. The synergistic use of GIS-based analysis and EO data can provide a useful tool for the design of natural hazards prevention policy at highly susceptible to risk landslide risk areas.

**Keywords:** landslide susceptibility; GIS-based analysis; Earth Observation; human risk mapping; Sperchios River basin

---

## 1. Introduction

Landslides are determined as the motion of debris, mass of rock, or earth down a slope due to gravity effect [1,2]. They are classified among the most dangerous and catastrophic natural hazards, being a significant threat to property and human life, moreover causing several indirect implications such as blocking of streams and aggregation of rivers, flash-flood occurrence, destruction of agricultural land, etc. [3–7]. In recent decades landslide risk has increased due to population growth, leading to the expansion of residential areas; establishment of new settlements towards steeper slopes and/or to areas of inappropriate foundation (bedrock, soil) in terms of structure and stability [2,8,9].

The definition of independent factors that reflect conditions prior to the landslide occurrence is of great significance. Data quality is particularly important in the field of landslide research, and highly related to the accuracy of the derivative results. The components responsible for landslide episodes variously and complexly interdepend. They can be distinguished into preparatory and triggering

mechanisms [10,11]. The preparatory mechanisms comprise general meteorological conditions, geomorphological processes, lithology (ground conditions and rock distribution), tectonic activity (the distance from the rupture plane and the seismic factor play a significant role on determining the landslide magnitude), vegetation cover, etc. The triggering mechanisms include a variety of external factors, such as intensive rainfalls, earthquakes (seismic-triggered landslides are classified as coherent, disrupted, and lateral spreads), speedy river erosion, landscape processes, and anthropogenic activities (deforestation and road construction in steep mountainous areas, uncontrollable irrigation, etc.) [12–17]. The designation of landslide risk areas can be achieved in one (or more) of the following ways: (a) landslide inventory maps, that display (at least) the geographical distribution of past events, followed by associated databases of landslide and terrain properties; (b) landslide susceptibility (LS) maps that refer to the tendency of an area to landslide occurrence—they depict the possibility of occurrence of a landslide event of a specific type at a particular place (where); (c) landslide hazard maps, which define the likelihood of occurrence of a potentially damaging landslide that may take place within a given area and period of time (where, how often, and how large)—this concept contains both spatial and temporal dimensions [8,18,19]; and (d) landslide risk maps that show potential damage or losses to individuals, infrastructure and property [20].

A landslide hazard map demonstrates regions susceptible to landslides by taking into account the phenomenon's causative and triggering factors (e.g., geomorphological, geological, and meteorological) with data concerning the past distribution of slope failures [12,21]. Landslide hazard assessment is crucial for the evaluation and management of natural disasters. It is also an essential stage for natural and urban planning in government policies worldwide [22,23]. Due to the potential lack of past landslides records, hazard estimation is not always possible [18,24]. Hazard assessment remains the only way to overcome this problem, identifying the place of potential landslides in a zone based on a set of terrain features [25]. Landslide risk associates the probability information from a landslide hazard map, with an analysis of all probable effects, taking into consideration the vulnerability and exposure of the area, i.e., the level of population; property; economic activity, the state of people; infrastructure; accommodation; and other actual human assets situated in hazard-prone areas [26].

Hence, the primary step in evaluating landslide risk is defining LS [5,12,27,28]. The LS term is commonly used to describe the position of potential landslides at a given area, based on a set of landscape characteristics. LS analysis considers that future landslides might be triggered by the same conditioning parameters from which present and past landslides were triggered [25]. Although these approaches make available information on potentially unstable slopes, they do not provide direct information on landslide extent and frequency. There are three LS assessment methods, categorized as qualitative, quantitative and semi-quantitative [8,12,29–31]. Qualitative methods tend to be very subjective, since they depend on expert opinion, with the weight of the factors leading to vulnerability being evaluated based on expert knowledge and experience [30,32,33]. The elementary types of qualitative methods utilize landslide index to identify areas with analogous geological and geomorphologic features that are susceptible to landslides [12]. Quantitative methods produce numerical estimates among controlling factors and landslides. They are categorized into deterministic and statistical [34]. The former depend on slope stability studies, signified in terms of the safety factor [35]. The latter examine the historical connection concerning landslide-controlling factors and the allocation of landslides, involving analytic bivariate, logistics regression, fuzzy logic, neural network, etc. Throughout the last decades, quantitative procedures have been applied for LS zonation researches in different areas [2,7,8,12,23,25,33,34]. Additionally, qualitative methodologies that operate in weighting and rating procedures are called semi-quantitative techniques [7,24]. These methodologies are the analytic hierarchy process (AHP) [2,5,13,26,36–40] and the weighted linear combination (WLC) [24,41]. The WLC method includes a combination of several landslide factors. Each factor is divided into classes, the layers of which are thereafter weighted according to the significance of each feature relative to the remaining ones. Finally, all the derived maps are overlaid in order to construct the final LS map [11,12].

Numerous LS mapping methodologies have been presented over the last decades [2,8]. Their aim is to categorize the different parts of the land surface according to the grade of actual or potential landslide susceptibility [5]. The new perspectives offered by earth observation and geographic information systems (GIS) technologies, provide the opportunity of an easy and integrated way to collect and process a variety of spatial data, necessary for LS estimation. GIS-based and remote sensing techniques have proven to be excellent tools in the spatial analysis of terrain factors for LS zonation [2,27,31,40,42–45].

One of the most significant variables affecting landslide susceptibility, apart from geology, weather conditions, and land cover, is topography (i.e., slope angle, aspect, and curvature) [46]. Therefore, the selection of a proper digital elevation model (DEM) is essential in LS assessment [47]. Japan's Advanced Spaceborne Thermal Emission Radiometer (ASTER) of Terra satellite makes available valuable geomorphic for landslide modelling, through the potential development of accurate and detailed DEM [48]. Likewise, the innovative Sentinel 2 (S2) mission of the European Space Agency Copernicus program comprises two identical satellites in the same orbit, 180° apart. Together they cover all Earth's surface, providing imagery of high spatial and temporal resolution (10 m and 5 day cycle, at the equator, respectively). Given its Multi-Spectral Imager (MSI) sensor, it operates on 13 different spectral bands covering visible (VIS, 4 bands, 1-2-3-4), near infrared (NIR, 5 bands, 5-6-7-8-8A), and short wave infrared (SWIR, 4 bands, 9-10-11-12) part of electromagnetic spectrum, having a wide swath coverage (290 km width). This satellite imagery is bringing land monitoring to a completely new level as a unique tool in the monitoring of vegetation conditions and classification. Land cover type and changes perform a crucial role in landslide appearance as they are an essential factor for stabilizing slopes [11,49].

The study objective is to identify the impact of the several natural and anthropogenic factors on LS at the Sperchios River basin, Central Greece, an area highly vulnerable to such phenomena. By adopting the analytic hierarchy process (AHP) for a weighted linear combination (WLC) approach, eleven evaluation parameters including lithology, slope angle, slope curvature, slope aspect, relative relief, land use/cover, soil depth, rainfall distribution, proximity to faults, and distance to roads and streams were initially selected as evaluation factors. All layers were subsequently converted to a raster layout, and a geospatial database was developed. Finally, the potential landslide risk of the area was estimated, considering the anthropogenic exposure (presence and activities) in the area. The study innovates by employing the applicability and efficiency of two state-of-the-art technologies, such as GIS and EO to competently collect process and incorporate the various spatial data necessary, complemented by well-established methods (AHP–WLC), for landslide susceptibility assessment and risk delineation.

## 2. Study Area

The study area is the Basin of the Sperchios River in Central Greece. It is located between 38°44′ and 39°05′ N lat., and 21°50′ and 22°45′ E long., occupying an area of approximately 1830 km$^2$ (Figure 1). The River's western and southern flanks comprise high elevations and steep slopes (Mt. Tymphristos 2315 m and Mt. Oiti 2152 m), contrary to its northern flanks where lower elevations (Mt. Othrys 1720 m) and milder slopes are met [50–53]. Mean annual precipitation for the period 1980–2010 was measured as 788 mm, considering the records of eight pluviometric stations [52].

The climatic of the study area is characterized as Csa (Mediterranean with hot summer) according to Köppen [54] indicating a dry warm period and a wet period distributed from late autumn to early spring. The mountainous areas of the studied basin are anticipated to form colder and wet conditions in terms of precipitation. Moreover, the topographic relief of the area enhances the formation of orographic precipitation during the year period.

According to Ferrière (1977) [55], there are three distinct lithological units in Sperchios basin, i.e., (i) the western part, dominated of Paleocene-Eocene flysch of the East Pindus and Parnassos geotectonic zones, including alternating beds of argillite-siltstone-fine conglomerate and intercalations of shale; (ii) the southeastern part, consisting of Middle Triassic-Jurassic massive dolomites and

limestones of the Pelagonian zone; Upper Cretaceous flysch consisting of coarse sandstone alternating with shale and sandy marl belonging to the Boeotia zone; and Upper Cretaceous thickly-bedded limestones and Eocene flysch, composed of sandstone, clay and marl of the Parnassos geotectonic zone; and (iii) the north and northeastern part (of utterly different lithological composition) with the presence of an ophiolitic complex in a shale-chert formation includes shale, chert, and limestone with ultra-mafic and mafic igneous rocks peridotites, dunites, pyroxenites, gabbros, serpentinites, diavases, dolerites, and metamorphosed green phyllite and schist; belonging to the Maliakos (Sub-Pelagonian) zone. The central and lower (elevations < 500 m) part of the watershed is occupied by Neogene and Quaternary unconsolidated deposits [50,56–58].

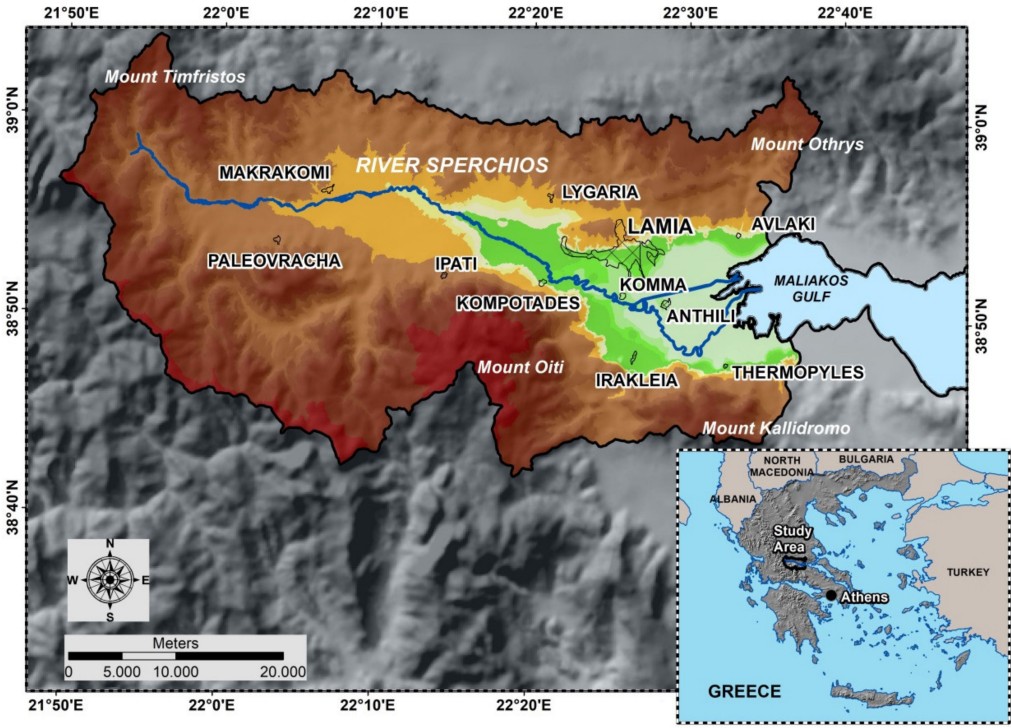

**Figure 1.** Location and geomorphological characteristics of the study area.

### 2.1. Landslides

In Greece, numerous landslide events have taken place over the years, especially at its western part, due to the effect of the presence of several preparatory factors such as the geological formations and the steep slopes (high relief energy). Moreover, the intense folding and joint tectonics, and the extreme weather conditions have favored the occurrence of significant landslide events [19,59–62].

According to the landslide hazard zonation map of Greece, the western regions of the Sperchios basin exhibit the highest frequency of landslides events (Figure 2a) [19,61]. The main preparatory factors of the landslide mechanism comprise the vulnerable geological formations (flysch) and the steep slopes, while the triggering factors are the extreme rainfall events and the ongoing tectonic activity [11,48]. Several landslide phenomena have occurred at the highly vulnerable western and southwestern parts of the catchment, mainly due to the extended presence of flysch formations (43% of the entire basin area). Such formations are related as much as 70–80% to the landslide's occurrence in Greece [62,63]. Apart from the bedrock characteristics, intense morphological relief with steep slopes, dense drainage network and deep valleys, and extensive human activities, such as relatively dense road network, and arable-irrigated areas are also responsible for landslide manifestations in the Sperchios watershed [12]. In the village Plaio Mikro Chorio, an area with similar geological and geomorphological characteristics, which is located approximately 30 km away from the basin's western edge, one of the deadliest landslides in the history of modern Greek state, occurred on the 13 January 1963. The landslide

destroyed the largest part of the settlement with severe consequences, causing 13 casualties [64]. Contrary, at the south-central and eastern parts (Neogene and calcareous formations), the few events that have been taken place are mainly related mainly to rock falls [19,52,58,65–67] (Figure 2b).

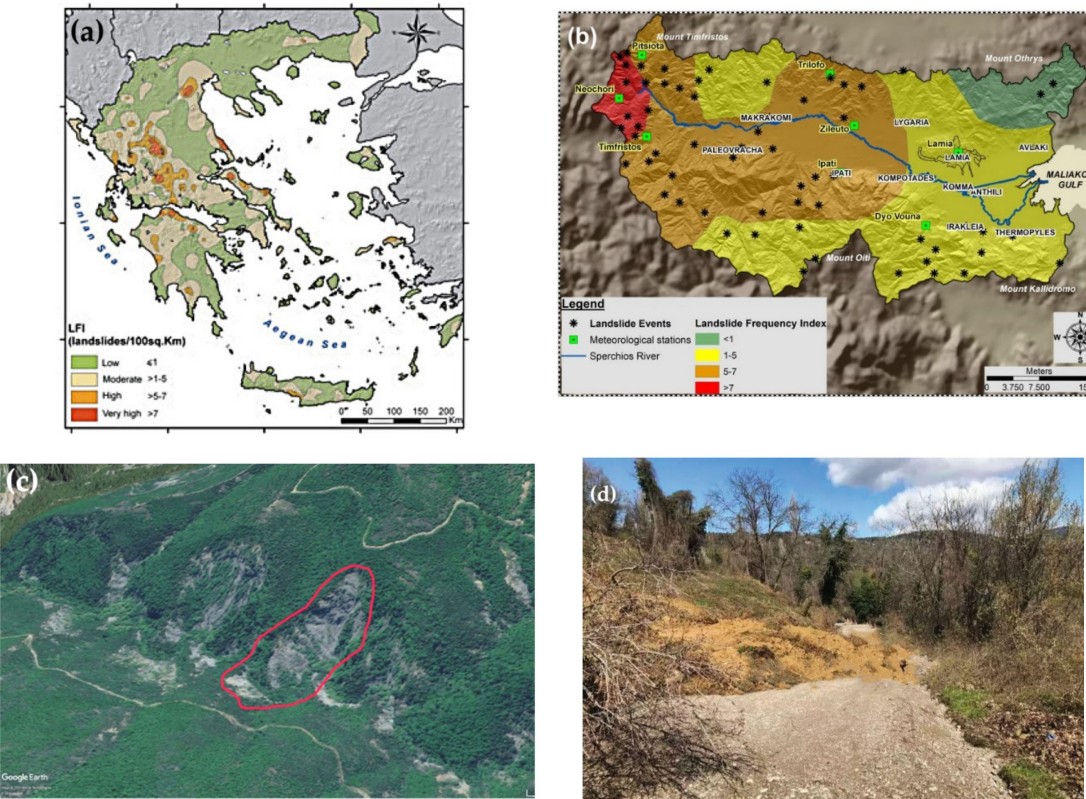

**Figure 2.** (**a**) The Landslide Frequency Index (LFI-landslides/100 km$^2$) map of Greece [16,56], (**b**) LFI of Sperchios river basin and the points of landslide events constitute the inventory map of the area (**c**) detection of past landslide event in Google Earth (red line), and (**d**) landslide detection during the fieldwork.

### 2.2. Inventory Map

A landslide inventory map provides valuable information on the spatial distribution of landslide events. The map is critical for examining the correlation among an event and its contributory factors.

The landslide inventory map of the study area was developed through the following stages: (a) collection of landslides historical information from previous studies (Figure 2a,b), (b) detection of landslides in very high resolution aerial photos and Google Earth satellite images (Figure 2c), and (c) fieldwork for the mapping of recent events (Figure 2d). Fifty-nine (59) landslide sites in total were gathered throughout the study area. Based on random sampling, the new landslide inventory map was separated into two datasets: 70% (40 landslide spots) for the triggering factors classification and weighting, and 30% (19 landslide spots) for the validation process, respectively [11,27].

## 3. Materials and Methods

### 3.1. Dataset

The different types of datasets utilized in the present study comprise (i) geological maps from the Institute of Geology and Mineral Exploration (IGME) at 1:50,000 scale [68]; (ii) topographic maps from the Hellenic Military Geographical Service at a scale of 1:50,000 [69]; (iii) satellite imagery (Sentinel 2 scenes); (iv) digital elevation model (DEM) of the area deriving from Terra/ASTER satellite data, level 1A, acquired on 30 July 2003 [70]; (v) precipitation records from eight (8) meteorological

stations (Figure 2b), overseen by the Hellenic National Meteorological Service (1 station), the Ministry of Environment, Energy, and Climate Change (4 stations and the Public Power Cooperation (3 stations); (vi) soil maps from the Directorates of Forests (Ministry of Agriculture) [71]; and (vii) population demographic characteristics from the Hellenic Statistical Authority [72].

More specifically, the geological maps were used to depict the basin's lithological and structural units, and the S2 images to delineate and supplement the lineaments-possible faults (derived from the geological maps) and extract the land cover types. The ASTER DEM (15 m) along with the topographic maps (20-m contour interval) was used to develop a detailed DEM, and to describe local topography (slope angle, slope aspect, slope curvature, and relative relief) and stream network. Finally, the precipitation records served as means to estimate mean annual precipitation for the period of 1971–2010 [73]. The data were incorporated towards compiling the final LS map. The geospatial database manipulation and the S2 and Aster satellite data processing were accomplished by utilizing ArcGIS (Environmental Systems Research Institute, Redlands, CA, USA) and ENVI (Harris Geospatial Solutions, Boulder, CO, USA) software, respectively. Moreover, the data pre-processing and a part of workflow's automation was made via R-Language scripts [74].

### 3.2. Methodology

The accuracy of susceptibility mapping improves when all landslide-controlling factors are incorporated in the analytical procedure, which is often hard to achieve, given the difficulty to acquire detailed data. Thus, analyses mainly depend on physical and anthropogenic factors.

The study's first task involved the selection of the landslide controlling factors, based on expert knowledge, extended field observations, literature review, and collection of available landslide historical data. Eleven factors were eventually selected, i.e., geological characteristics, such as (i) lithology, (ii) proximity to major faults; geomorphological features, such as (iii) slope angle, (iv) slope aspect, (v) slope curvature, (vi) relative relief, (vii) drainage network, (viii) soil depth; and anthropogenic factors and weather information, such as (ix) road network, (x) land use/cover, and (xi) rainfall distribution [12,19]. The next step was to assign weights and rank values per factor (i.e., raster layer; pixel size 20 × 20 m) and factor class (within each layer), respectively. Two methods were used for the LS index calculation, the simple and the geometric mean (GM). Regarding the first method, the weight calculation was based on the analytic hierarchy process [36], providing an objective approach to such an assignment [13,27]. Regarding the second method, the rating of the classes considers the most stable regions of each parameter and classifies them with zero values. The landslide susceptibility indices (LSI) were developed utilizing the raster layers and their corresponding weights using the two methods [12–14]. Finally, the landslide risk map was produced, through the evaluation of landslide hazard map impact on anthropogenic exposure and vulnerability of the study area. The methodology applied is schematically presented in Figure 3 and is analyzed in the following sections.

### 3.2.1. Digital Elevation Model from ASTER Data

A strong advantage of the along-track mode of the ASTER data acquisition system is that it provides stereo pairs of images acquired a few seconds separately under the same conditions that are appropriate for DEM creation by stereo-correlation techniques [48,75].

The stereo pair of the near-infrared backward and nadir images (3N and 3B) was used for the DEM creation. The synchronized along-track stereo-data acquisition offers a robust benefit in terms of radiometric differences contrasted. Stereo-correlation is a computational and statistical method used to obtain a DEM from the registered images of stereo pairs. For the photogrammetric matching rational polynomial coefficients (RPC) of the images along with well-distributed ground control points were applied. As a result, a very precise DEM with a 15 m spatial resolution was produced.

The ASTER DEM was compared to a DEM developed from 1:50,000 topographic maps' digitized contours. The visual comparison of the DEMs led to the detection of errors and significant alterations in the elevation data of the DEM deriving from the digitized contours; thus, it required updating.

The vertical accuracy of the ASTER elevation data was checked using three hundred points of known elevation with the root mean square error (RMSE)-z being was ± 3.2 m. Therefore, the accuracy of the ASTER DEM is considered appropriate for updating the DEM deriving from the 1:50,000 topographic maps [75].

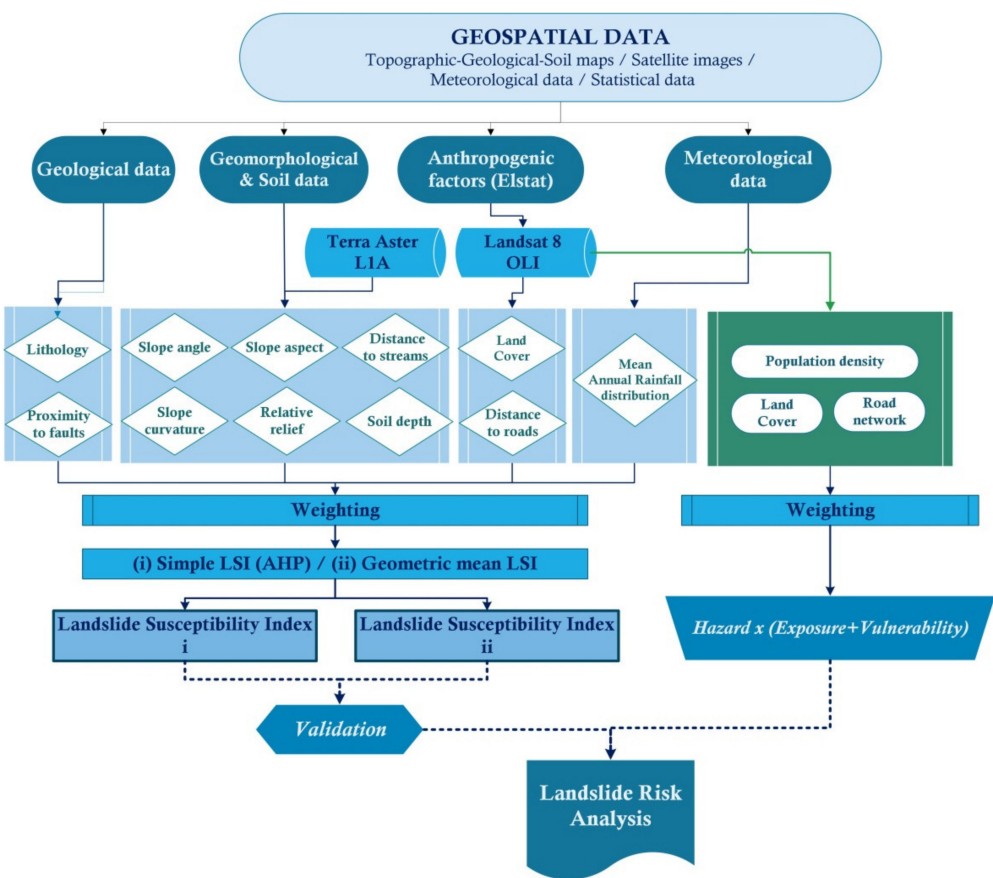

**Figure 3.** Methodology flowchart.

### 3.2.2. Land Cover Classification from Sentinel Data

Land cover classification based on S2 images has progressed over the last five decades, having been used for the identification of different land cover types through various classification methods applied on EO data [76]. S2 MSI data belongs to the new generation of satellites with improvements that are principally characterized by enhanced spectral, spatial, and radiometric resolution and were utilized for land cover mapping [77]. Four cloud-free, geometrically, and atmospherically corrected images, of the year 2019 (one for every season), were acquired, to distinguish the different land cover types (Figure 4). The images were acquired free of charge via the European Space Agency (ESA) portal (https://scihub.copernicus.eu/) [11,78].

The land cover map derived from the supervised classification, utilizing the maximum likelihood method with 440 ground control samples (collected through field survey and orthophoto maps). The data samples were divided into the training (70%) and the validation data set (30%). Using the first (training) set, and twelve classes, a two-step classification procedure was followed. The first step comprised a broad distinction between general land cover types (forest, water, built-up classes, etc.). In the second step, the general LC types were further classified into summer and winter arable crops, and different kinds of forest types (coniferous, broad-leaved, etc.). The accuracy assessment of the classification map was accomplished using the second validation set. The results were assessed using overall accuracy (OA), and the Kappa coefficient [79,80]. The classification OA and Kappa coefficient were estimated at 95.8% (295 correctly classified parcels divided to 308 total reference parcels) and 0.88,

respectively, indicating a highly accurate result. Finally, post-classification filtering was performed to remove high-frequency deviations and misclassified pixels.

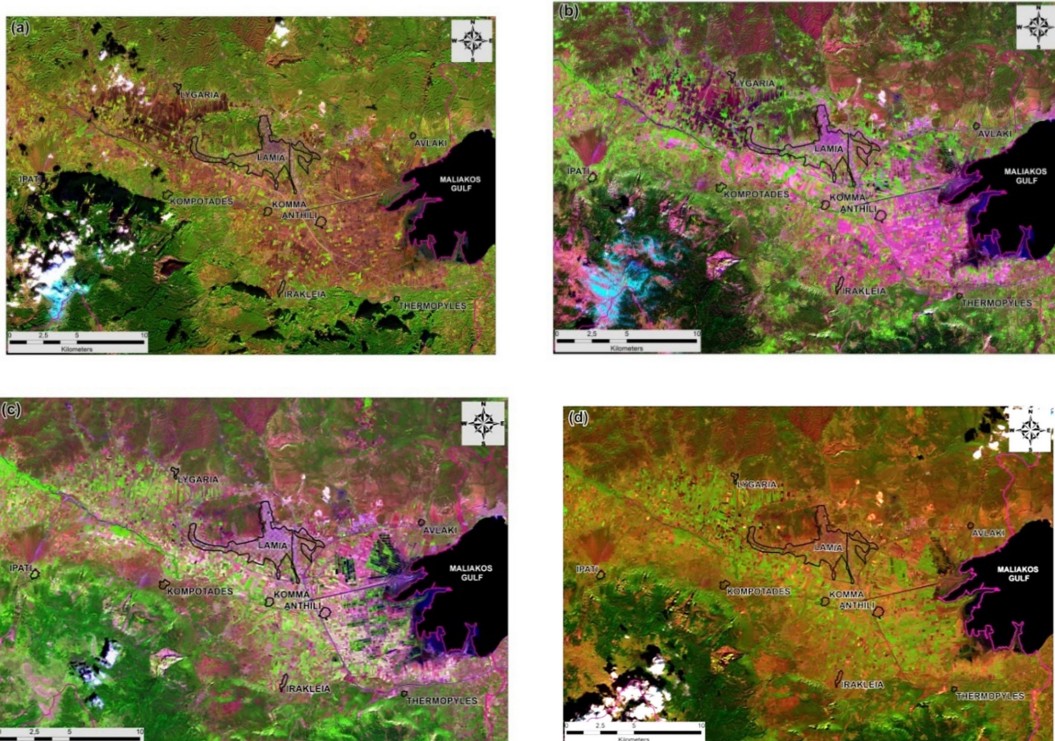

**Figure 4.** Sentinel-2 images acquired on (**a**) December 2018 (winter image), (**b**) April 2019, (**c**) July 2019, and (**d**) September 2019, and used for the natural vegetation and crop classification procedure.

### 3.2.3. Data Layers Formation

The factors chosen for the estimation of landslide susceptibility in the study area were incorporated into a geospatial database. All thematic layers were georeferenced to the Greek Geodetic Reference System (GGRS'87). The rating classes of each factor were determined based on the first dataset of the inventory map maintained for their classification and weighting, along with previous researches [12,19] and our experience on the local conditions of the study area.

Lithology: Lithology is one of the most critical controlling factors of landslides since each lithological unit displays different susceptibility rates. Several researchers [5,28,62,65] emphasized the role of geology on slope stability. The lithology layer derived by digitizing the IGME (1957–1991) geological maps. The bedrock formations were classified into five general categories concerning LS, i.e., (a) limestones and marbles, (b) Neogene sediments, (c) schists and ophiolites, (d) alluvial deposits and debris, and (e) flysch (Figure 5a—Table 1).

Land Cover: Vegetation is a potential driver of LS. Some land cover types, e.g., forest trees with strong and extensive rooting system, enhance slope stability through their effect on the soil's hydrological and mechanical attributes [12,81]. The land use/cover data deriving from L8 data was reclassified into 7 classes based on their effect on LS, i.e., (a) coniferous and broadleaves forest; (b) deciduous forest; (c) tree crops; (d) urban areas; (e) arable land; (f) vineyards and nude soils; and (g) pasture land, grassland, and burnt areas (Figure 5b—Table 1) [82].

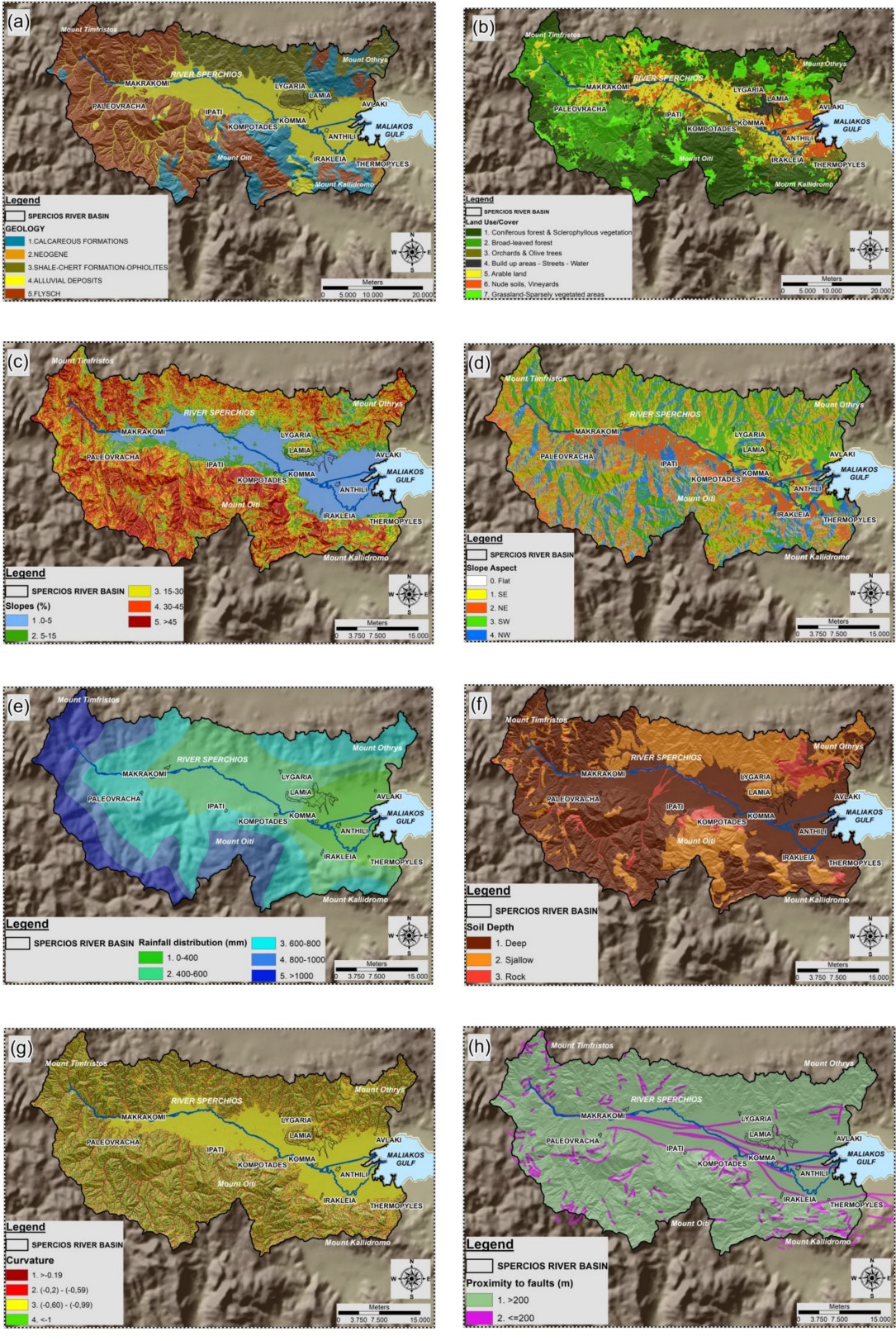

**Figure 5.** *Cont.*

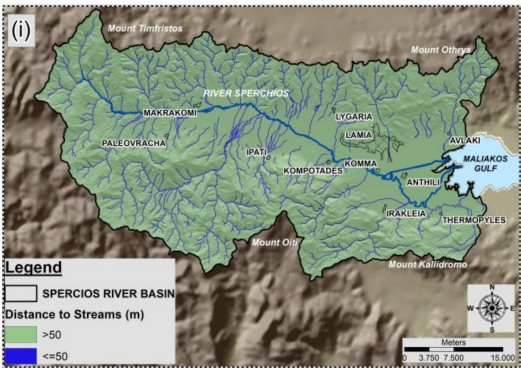
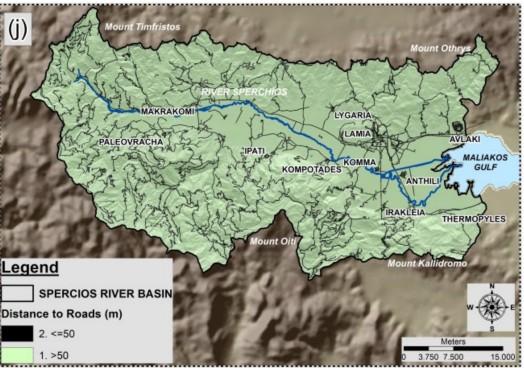
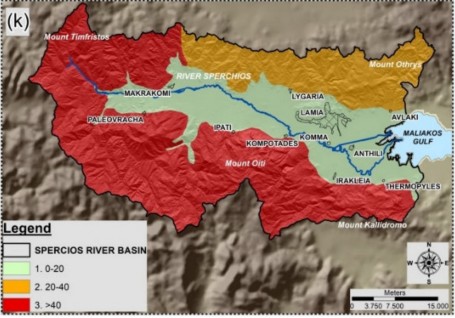

**Figure 5.** The thematic raster maps of the eleven (11) factors, used for the estimation of Landslide Susceptibility of Sperchios River basin: (**a**) Geological map, (**b**) Land use/cover map, (**c**) Slope map, (**d**) Slope aspect map, (**e**) Rainfall distribution map, (**f**) Soil depth map, (**g**) Curvature map, (**h**) Proximity to faults, (**i**) Distance to streams, (**j**) Distance to roads, and (**k**) Relative relief map.

Slope angle: Slope angle and geometry control slope stability and affect surface runoff, in conjunction to soil permeability and saturation capacity [5,7,12,33,40,83]. Gentle (low gradient) slopes are expected to display less LS than the steep ones, as in the case of the study area (although in some other cases surface landslides occur on low slopes in recent soils). Slope angle was modelled using a grid-based DEM, developed from topographic maps contour lines (20 m intervals) validated from the ASTER-derived DEM [28]. The slope angle was classified into five classes based on their effect on LS, as (a) very gentle slopes; 0–5%, (b) gentle slopes; 5–15%, (c) moderate steep slopes; 15–30%, (d) steep slopes; 30–45%, and (e) very steep slopes; >45% (Figure 5c—Table 1).

Slope aspect: The slope aspect defines the direction of slopes. Slope orientation indirectly influences landslides as it controls the exposure of slopes to various climatic features, such as rainfall, sun, and by extension, vegetation cover [84]. Slope aspect was grouped into five classes, as (a) 0–90° as NE, (b) 90–180° as SE, (c) 180–270° as SW, (d) 270–360° as NW, and (e) flat regions (Figure 5d—Table 1).

Mean Annual Rainfall distribution (MARd): Rainfall distribution affects overland water runoff volume and soil moisture. In the present study, the data of eight meteorological stations were used, covering 40 years (1971–2010). Precipitation was spatially distributed based on its mean annual values, utilizing the co-kriging interpolation method, using as covariate the basin elevation, as described in detail by Soulis et al. (2018) [85]. Thus, MARd was classified into five classes, as (a) 0–400 mm, (b) 400–600 mm, (c) 600–800 mm, (d) 800–1000 mm, and (e) >1000 mm (Figure 5e—Table 1) [11,48].

Soil depth: Soil depth affects soil percolation and the shape of the slope. Three classes were developed, based on the thickness of the top-soil layer, as (a) deep soils (>80 cm), (b) shallow soils (20–80 cm) and (c) rock (without soil cover or <20 cm) (Figure 5f—Table 1) [48].

**Table 1.** Classification of the landslide controlling factors and categorization into rating classes according to their significance on Landslide Susceptibility.

| Factor | Class | Class Value Rating (i) | Class Value Ratting (ii) |
|---|---|---|---|
| Slope Angle (%) | 0–5 | 1 | 0 |
| | 5–15 | 2 | 2 |
| | 15–30 | 3 | 3 |
| | 30–45 | 4 | 4 |
| | >45 | 5 | 8 |
| Lithology | Carbonate formations | 1 | 0 |
| | Neogene | 2 | 1 |
| | Schists-Ophiolites | 3 | 2 |
| | Alluvial dep.-Debris | 4 | 3 |
| | Flysch | 5 | 6 |
| Land Use/Cover | Coniferous forests | 1 | 0 |
| | Deciduous Forests | 2 | 0 |
| | Orchards-Olives | 3 | 3 |
| | Urban-Roads | 4 | 4 |
| | Arable land | 5 | 5 |
| | Nude soil and rocks | 6 | 8 |
| | Pastures | 7 | 8 |
| Slope Aspect | Flat areas | 0 | 0 |
| | SE | 1 | 1 |
| | NE | 2 | 2 |
| | SW | 3 | 3 |
| | NW | 4 | 4 |
| Relative Relief (m) | <20 | 1 | 0 |
| | 20–40 | 2 | 2 |
| | >40 | 3 | 3 |
| Slope Curvature ($m^{-1}$) | >−0.19 | 1 | 0 |
| | (−0.2)–(−0.59) | 2 | 2 |
| | (−0.6)–(−0.99) | 3 | 3 |
| | ≤−1 | 4 | 4 |
| Soil Depth (m) | Deep | 1 | 0 |
| | Shallow | 2 | 2 |
| | Rock | 3 | 3 |
| Distance to streams (m) | >50 | 1 | 0 |
| | ≤50 | 2 | 2 |
| Proximity to faults (m) | >200 | 1 | 0 |
| | ≤200 | 2 | 2 |
| Distance to roads (m) | >50 | 1 | 0 |
| | ≤50 | 2 | 2 |
| Rainfall (mm) | 0–400 | 1 | 0 |
| | 400–600 | 2 | 2 |
| | 600–800 | 3 | 3 |
| | 800–1000 | 4 | 4 |
| | >1000 | 5 | 8 |

Slope Curvature: Slope curvature is a measure of a slope's morphology and topography. Positive values express upward curved surfaces, while negative values upward concave surfaces [30,86]. The lower negative values indicate greater LS. The classification of the slope curvature map comprises the following classes (a) ≥−0.19, (b) −0.2 to −0.59, (c) −0.60 to −0.99, and (d) ≤−1 (Figure 5g—Table 1).

Proximity to faults: Faulting earthquakes are considered as an important triggering factor of landslide events [16,17,87]. Moreover, contiguity to tectonic structures creates the possibility of a landslide phenomenon since erosion processes and water flow along a crack might be caused. For the fractures (both active and non-active faults), a buffer zone of 200 m width was developed, and the area was divided into two sections, i.e., inside and outside the zone (Figure 5h—Table 1) [11,48].

Distance to streams: Vicinity to streams is an important controlling factor of landslides since it can cause significant erosion processes (gully erosion, i.e., Gorgopotamos gorge) [5,88]. Third or higher-order streams (according to Strahler's classification) were selected and processed by creating a 50 m buffer zone. Thus, the basin area was divided into two sections, i.e., within and outside the zone (Figure 5i—Table 1).

Distance to roads: During the development of road networks, extensive excavations, and deforestation; removal of natural vegetation takes place quite often. Such actions, in conjunction with other natural factors (e.g., intense precipitation events), can cause significant landslide phenomena. A buffer zone of 50 m around the main road network of the study area was set, dividing it into two sections, i.e., within and outside the region (Figure 5j—Table 1) [7,11,48].

Relative relief: Relative relief portrays the absolute maximum difference in elevation at a specific point. Comparing slopes with identical geo-mechanical and geometrical parameters, the ones with the greater elevation deficit are more susceptible to landslides. In such landscapes, higher runoff and lower infiltration rates are expected [28,89]. Relative relief was created by using a unit area of $100 \times 100$ m and classified into 3 classes, as (a) 0–20, (b) 20–40, and (c) >40 (Figure 5k—Table 1).

### 3.2.4. Analytic Hierarchy Process

In AHP, all factors are compared pairwise in terms of the intensity of their importance using a continuous 1 to 9 point scale according to Saaty [90].

The nine-point scale of preference between two parameters in AHP is analyzed as follows 1: Equal importance i.e., two factors contribute equally to the objective, 3: Moderate prevalence of one over another i.e., experience and judgment slightly to moderately favor one factor over another, 5: Strong or essential prevalence i.e., experience and judgment strongly or essentially favor one factor over another, 7: Very strong or demonstrated prevalence i.e., a factor is strongly favored over another and its dominance is shown in practice, 9: Extremely high prevalence i.e., the evidence of favoring one factor over another is of the highest degree possible of an affirmation, 2; 4; 6; 8: Intermediate values i.e., reciprocals for inverse comparison, used to represent compromises between the preferences in weights 1, 3, 5, 7 and 9; Opposites i.e., used for inverse comparison.

Thus, the pairwise comparison matrix was created to calculate factor weights in AHP (Table 2). The diagonal boxes are assigned with a unit value, while the boxes in the upper and lower halves are symmetrical to one another and the corresponding values, and consequently reciprocal with each other. When the factor on the vertical axis is more important than the factor on the horizontal one, the values vary between 1 and 9. At the same time, and contrary to the previous notion, the value varies between the 1/2 and 1/9 reciprocals.

An essential feature of the AHP is that allows defining rating inconsistencies by the consistency index (*CI*), which is used defined by Equation (1) [36,91]:

$$CI = \frac{\lambda_{max} - N}{N - 1} \qquad (1)$$

where $\lambda max$ = the largest eigenvalue and $N$ is the order of comparison matrix.

**Table 2.** Pair-wise comparison matrix and weights (normalized principal eigenvector) for landslide causative factors, derived from the analytic hierarchy process (AHP) method application.

| | (i) | (ii) | (iii) | (iv) | (v) | (vi) | (vii) | (viii) | (ix) | (x) | (xi) | Weights |
|---|---|---|---|---|---|---|---|---|---|---|---|---|
| Geology (i) | 1 | 2 | 2 | 4 | 4 | 4 | 2 | 5 | 6 | 7 | 8 | 0.227 |
| Slope Gradient (ii) | 1/2 | 1 | 1 | 2 | 3 | 3 | 3 | 3 | 6 | 7 | 8 | 0.155 |
| Rainfall Distribution (iii) | 1/2 | 1 | 1 | 1 | 3 | 3 | 3 | 3 | 5 | 6 | 7 | 0.140 |
| Land Use/Cover (iv) | 1/4 | 1/2 | 1 | 1 | 3 | 3 | 3 | 3 | 6 | 7 | 8 | 0.133 |
| Slope Curvature (v) | 1/4 | 1/3 | 1/3 | 1/3 | 1 | 3 | 3 | 3 | 6 | 7 | 8 | 0.10 |
| Slope Aspect (vi) | 1/4 | 1/3 | 1/3 | 1/3 | 1/3 | 1 | 3 | 3 | 6 | 7 | 8 | 0.081 |
| Soil Depth (vii) | 1/2 | 1/3 | 1/3 | 1/3 | 1/3 | 1/3 | 1 | 3 | 5 | 6 | 7 | 0.067 |
| Relative Relief (viii) | 1/5 | 1/3 | 1/3 | 1/3 | 1/3 | 1/3 | 1/3 | 1 | 3 | 5 | 6 | 0.044 |
| Proximity to faults (ix) | 1/6 | 1/6 | 1/5 | 1/6 | 1/6 | 1/6 | 1/5 | 1/4 | 1 | 4 | 3 | 0.024 |
| Distance to Rivers (x) | 1/7 | 1/7 | 1/6 | 1/7 | 1/7 | 1/7 | 1/6 | 1/5 | 1/4 | 1 | 2 | 0.015 |
| Distance to Roads (xi) | 1/8 | 1/8 | 1/7 | 1/8 | 1/8 | 1/8 | 1/7 | 1/6 | 1/3 | 1/2 | 1 | 0.012 |
| | | | | | | CR = 0.072 | | | | | | |

Saaty (1980) [36] developed an average random consistency index (RI) for different matrix orders and defined the consistency ratio (CR) as the ratio between the consistency index (*CI*) and the random consistency index (RI). The CR depends on the number of parameters. In case that CR is greater than 0.1, the comparison matrix is inconsistent and should be revised. In the present case, the CR value is less than 0.10 (0.08), which clarifies that the preferences used to create the comparison matrices are consistent [7,27,92].

### 3.2.5. Weighting of Parameters—Landslide Susceptibility Index

The various classes of the thematic layers were designated with the corresponding rating values inputted as attribute records in a GIS environment, and a raster map was developed for every data layer. Subsequently, the reclassified raster layers of the eleven produced maps were used as input data for the LSI estimation. After the weights' assignment, all factors were combined using the raster calculator procedure. The equations deriving from the implementation of the two different techniques, the Simple LSI and the GM-based LSI method [12,93] are analyzed as follows.

(a) Simple LSI (i):

Simple LSI (i) was calculated by multiplying the assigned rate values (Table 1) of the raster layers with the corresponding weights deriving from the AHP method (Equation (2)),

$$\text{LSI (i)} = \sum_{i=1}^{n} weight_{AHP} \times \text{class rate (i)} \tag{2}$$

where *n* is the total number of data layers.

(ii) Geometric mean LSI (ii):

The geometric mean is a multiplication-based method that ends when a null (0) value arises and is useful for concealing areas characterized by factors considered unconnected (the most stable areas) to the potential incident. By implementing the zero values the GM excludes the generally stable areas from the estimation as it depicts the areas which are predominantly vulnerable to landslides. The GM is assessed by the following Equation (3) [12,91,94],

$$\text{LSI (ii)} = \text{GM} = \left( \prod_{i=1}^{n} class\ rate\ (ii) \right)^{1/n} \tag{3}$$

where n is the total number of data layers; in order to serve the mathematical calculations, the zero values were replaced with the value 0.001.

The final step of the process involves the LSI maps validation alongside with landslide events and their distribution, having derived from the inventory database, and supplemented with potential landslide locations. The latter were mapped utilizing satellite images, very high-resolution orthophotos, and field observations serving as ground truth (Figure 2a,b) [11,19,62,65]. The final maps of the already applied methods were also compared and validated in order to select the most suitable for the study area [95,96].

### 3.2.6. Validation

The two LSI maps were validated based on two methodological approaches. The first method makes a comparison with the inventory map of the historic landslide events that occurred in the study area. Therefore, using the dataset that "preserved" for validation purposes, a GIS-based statistical analysis was made. The second method was formulated by creating a fishnet (3000 × 3000 m/point) that comprises 202 points located in the basin, which were intersected with the two LSI maps and the landslide frequency index (Figure 2b). In this case, the validation, in this case, was made using the root mean square error (RMSE) (Equation (4)). RMSE calculates the predicting errors of the two indices or in other words, measures the difference between the value predicted by each method and the LFI map,

$$\text{RMSE} = \sqrt{\frac{1}{n} \sum_{i=1}^{n} \left(V_{\text{predicted}} - V_{\text{actual}}\right)^2} \tag{4}$$

where $n$ is the number of samples in the dataset used, $V_{\text{predicted}}$ is the value predicted from each method, and $V_{\text{actual}}$ is the actual value in the LFI map.

### 3.2.7. Landslide Risk Analysis

In recent decades, increasing population and expansion of settlements joint by the subsequent intensification of anthropogenic activities (such as cultivations and transportation network) over vulnerable areas have essentially increased the impact of natural disasters. In the Sperchios basin the population increased approximately by 13.3% during the last four decades (with a small decrease in the last decade), while the extent of the urban fabric increased by almost 19.9%. Moreover, other intense human activities like land cultivation have increased by approximately 5.1%, while several public construction projects (some of noteworthy scale), have taken place in the region, involving the reconstruction of the main National Highway and Railway, the construction of the E65 highway, and enhancement and expansion of numerous secondary road network.

The LS map was managed in order to initially assess the landslide hazard zoning (LHZ) of the study area. LHZ assigns the estimated frequency (i.e., annual probability) to potential landslides and can be expressed as the specific landslide type of a given volume [97,98]. The determination of landslide hazard range requires the detection of the vulnerable areas, and the calculation of the probability of landslide occurrence within a pre-defined period. Since this period is often challenging to be demarcated, landslide hazard is frequently represented by LS [99].

Landslide risk (LR) utilizes the hazard mapping results to assess the landslide probability of occurrence and its severity to individuals (annual probability of human loss), property (the annual value of property loss), and environmental assets. Hence, LR depends on temporal and spatial probability (hazard), the vulnerability to such phenomena, and the human exposure to danger [12,84,100,101]. LR can be estimated based on Equation (5):

$$\text{LR} = \text{Hazard} \times \text{Exposure} \times \text{Vulnerability} \tag{5}$$

Exposure was assessed by creating a map of human activity in the area. The map was based on a combination of three factors, population density, land use/cover, and the road network extent. These factors were classified, with their classes being ranked based on the exposure to risk (Table 3). Vulnerability signifies the degree of loss to a given component within the area affected by the landslide, and it is expressed on a scale of 0 (no loss) to 1 (total loss). For human life and property, the loss is the value of the damage concerning the probability that peoples can be affected by the natural disaster (landslide) [12,101].

**Table 3.** Classification of risk factors and categorization into rating classes according to their significance on Landslide Risk.

| Risk Factors | Class (Exposure) | Vulnerability | Class Rating | Importance |
|---|---|---|---|---|
| Population density | Low | Low to Moderate (0) | 1 | 10 |
| | Moderate | | 2 | |
| | High | High to Very High (1) | 3 | |
| | Very High | | 4 | |
| Land Use/Cover | Areas with low human intervention (Forests, pastures, etc.) | Low | 1 | 8 |
| | Cultivated areas | Moderate | 2 | |
| | Urban areas | High | 3 | |
| Road network | Distance to road >50 m | Low | 1 | 6 |
| | Distance to road ≤50 m | High | 2 | |

Subsequently, the region's vulnerability and the weights of the three exposure factors were calculated (Table 3). The highest values were attributed to areas where human presence and activity are intense. Therefore, the impact of landslides in the corresponding risk assessment is high (Table 3). After that, the "human exposure" map was created by reclassified the map into three categories of Low to Moderate, Moderate to High, and Very High anthropogenic presence (Figure 6). The percentage of each category in the basin was 38.02%, 35.17%, and 26.81%, respectively.

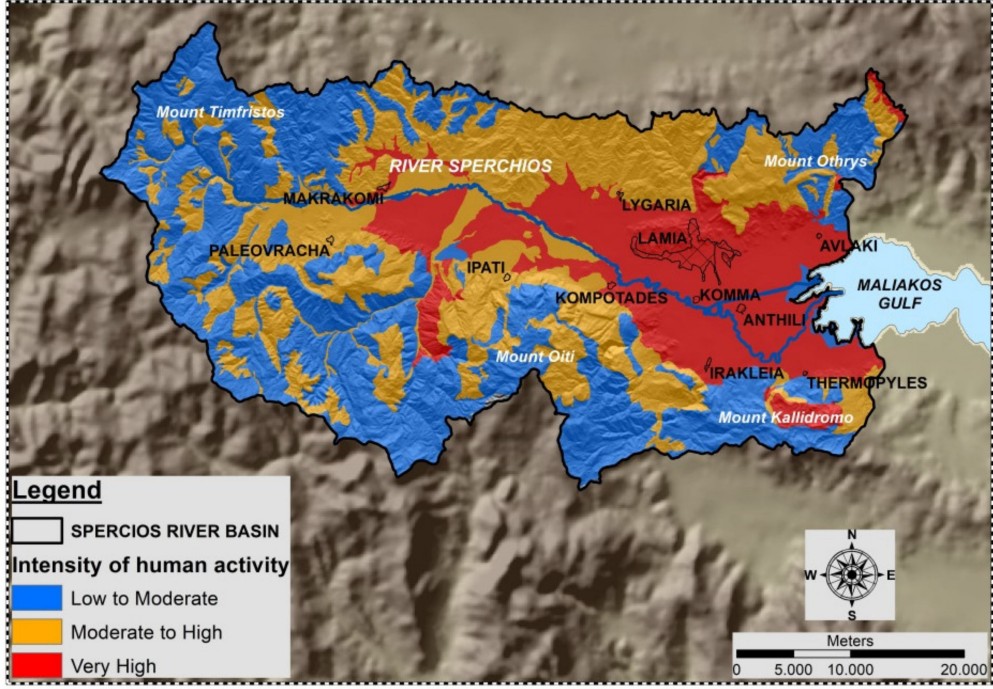

**Figure 6.** Representation of anthropogenic activity distribution in the Sperchios river basin.

## 4. Results and Discussion

From the statistical evaluation of the eleven selected factors related to landslide events, the distribution and correlation of past landslides to each factor was estimated. The analysis showed that in the geological map, 61.0% of past landslides are located on flysch formation (37 events), 13.6% on carbonate formations, and 13% on ophiolites (8 and 7 events, respectively). This high percentage of landslide events on flysch, which covers 42.8% of the basin, is due to its compound characteristics and especially to its composition, consisting of clayey or psammitic materials. This clayey, and usually thick soil layer, due to its high capacity to absorb water and then soften, or due to the existence of weathered basement rocks beneath the humus, creates potential slip surfaces (at the upper zone or between the weathered area and the bedrock), which are very susceptible to landslide occurrences. It was evident that the phenomenon is directly related to extreme weather events of high rainfall amounts that interact with the clayey flysch formation triggering numerous of landslide events. Likewise, very steep slopes lead to intensive instability conditions that contribute directly to the landslide phenomena. Slides that usually take place in the gentle slopes of the flysch mantle are typically quite shallow and take the form of a sheet of weathered zone sliding on a slip surface parallel to the ground [60,102,103]. Contrary, at the south-central and eastern parts where calcareous formations appear, the events are primarily related to rock falls.

60% of the study area is characterized by mean annual rainfall amounts higher than 600 mm. From these, 29% are greater than 800m, while the 44% falls on steep and very steep slopes, and 25% falls on moderate slopes. These characteristics and the fact that 39% of the basin's steep and very steep slopes, amounts are situated on flysch formation (where 38% of the mean annual rainfall amounts falls), generates large susceptible areas prone to fast weathering and gradual weakening, mostly triggered by prolonged or short intensive rainfall events, which result in many rotational slides and mudflows [52,104].

Furthermore, 74.5%, 43.1%, and 83.05% of the past landslide events occurred in forested areas, deep soils, and high relative relief, probably due to the fact that these characteristics mainly appears in the mountainous southern part which is also composed of flysch. This part of the basin exhibits intense tectonic activity (the basin is a graben which functions as a tectonic dipole, where the south part is lifted, and the north part is sunk), higher relief with very steep slopes (at a relatively small distance, an altitude difference of 300 m can be noted) and very high precipitation depths (>1000 mm) [52]. At the same time, the northern part displays milder topography with lower altitudes. Surprisingly, the number of landslides on slope aspects is almost equal to all directions, contrary to the fact that NW and SW aspects are more susceptible to landslide events in Greece [59]. The reason for this is probably the existence of all the above-mentioned characteristics in the SW part of the basin, which inevitably have a SE or NE orientation [52,60].

The correlation of landslide manifestations to faults, streams, and roads proximity, appears to be stronger to roads (27.1%), whereas the link is very weak (11.86%) to faults and streams (10.2%). Hence, these factors seem to have a meager contribution to landslides.

### 4.1. Landslide Susceptibility Indices

The LSI maps were classified into five categories, Very Low, Low, Moderate, High, and Very High susceptibility (Figure 7a,b). The generated maps were processed (dissolved and smoothed out using a 5 × 5 low-pass filter) to decrease the significance of possible misclassified cells. The percentage distribution of the two created LSI maps classes is demonstrated in Table 4. Higher LSI values indicate areas where the combination of factors is more likely to result in landslide events.

The GIS-based statistical analysis revealed that the most susceptible areas to landside manifestation (high and very high values) are placed at (a) slope angles with very high and high values approximately 66.17% for LSI (i) and 31.11% for LSI (ii), (b) flysch formation of about 72.33% for LSI (i) and 49.77% for LSI (ii), (c) mean annual precipitation > 1000 mm around 69.69% for LSI (i) and 34.41%, for LSI (ii), and (d) forest land cover approximately 59.21% and 48.62% for LSI (i) and LSI (ii), respectively.

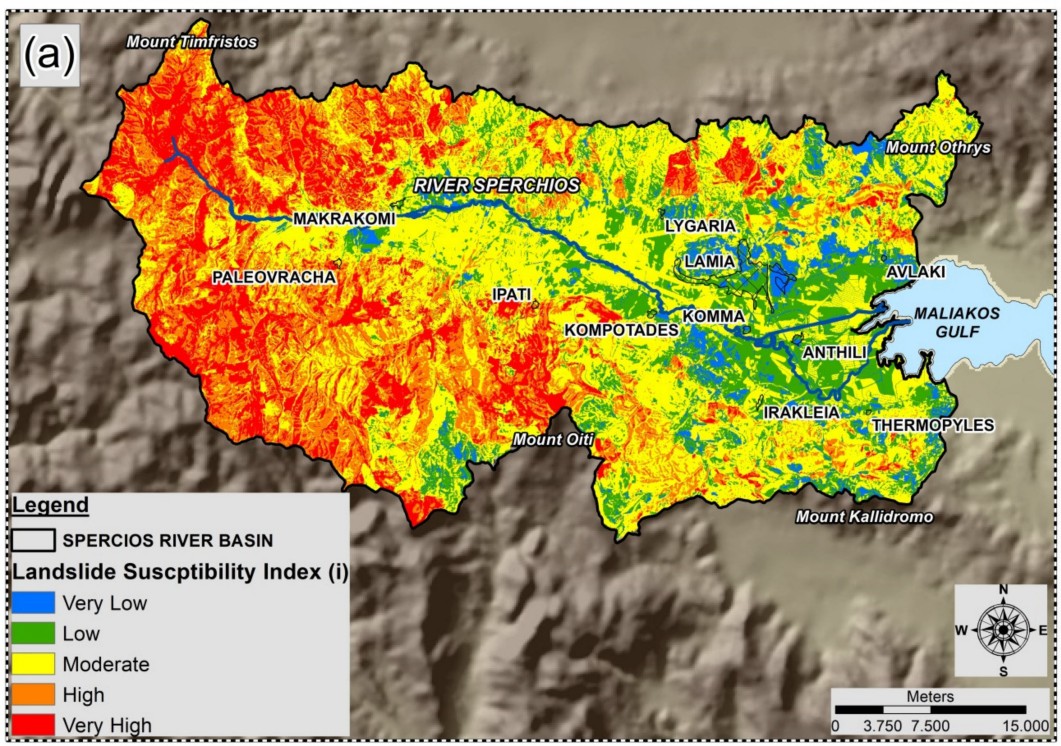

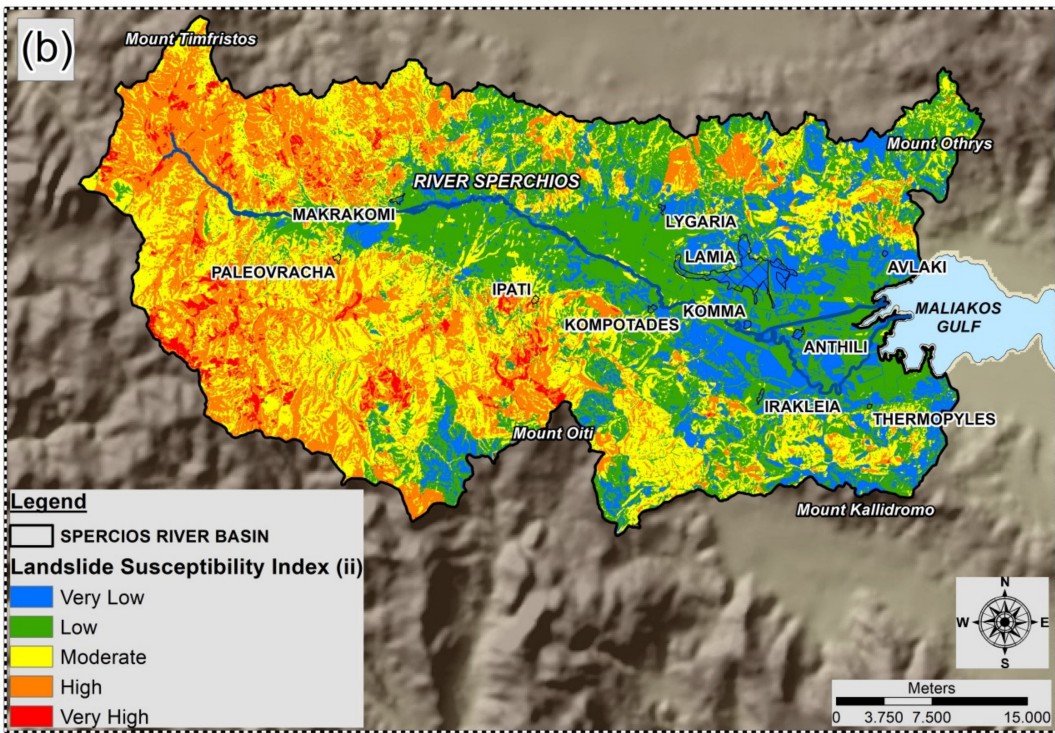

**Figure 7.** (**a**,**b**). The Landslide Susceptibility Indices (i) and (ii), after reclassifying the calculated values into five classes of potential susceptibility.

**Table 4.** Percentage distribution of the five classes of the two created landslide susceptibility (LSI) (i) and LSI (ii) maps, using the two different methods.

| Susceptibility Classes | Very Low (%) | Low (%) | Moderate (%) | High (%) | Very High (%) |
|:---:|:---:|:---:|:---:|:---:|:---:|
| LSI (i) | 4.52 | 16.03 | 43.09 | 21.00 | 15.35 |
| LSI (ii) | 14.34 | 30.39 | 29.79 | 23.18 | 2.31 |

It is apparent that the most sensitive areas (high and very high susceptibility) to landslides are mainly located in the western and southwestern parts of the basin, areas where flysch formations appear on steep slopes in conjunction with the highest rainfall amounts, and less in the lowland and coastal areas. These regions are dominated by flysch (capable of absorbing and retaining large amounts of water due to the formed clay-rich soils); moreover, they are characterized by intense and prolonged rainfalls, very steep slopes, and convex slope curvatures. Consequently, when the moisture amount in these soils exceeds a "moisture limit", these soils become extremely susceptible to landslides.

*4.2. Accuracy Assessment*

Using the first method, the validation analysis showed satisfactory results for LSI (i) since the overall matching acquires a value of 89.5% (Table 5). Contrary, the LSI (ii) showed a much lesser accuracy with a value of approximately 68.4% [11]. The RMSE indicated that the LSI (i) and LSI (ii) maps display a value of 0.21 and 0.43, respectively. The results of the method are perfect when RMSE values are equal to 0 [105,106]. Apparently, from the two validation approaches, can be determined that the LSI (i) method provides a more accurate assessment of landslide susceptibility in the study area.

**Table 5.** Confusion matrix of LS map validation.

| Validation Sample | | Target Class (Observed) | |
|:---:|:---:|:---:|:---:|
| | | Susceptible Areas (Moderate, High, Very High Classes) | No Susceptible Areas (Low, Very Low Classes) |
| LSI (i) | Landslide areas | 17 | 2 |
| | Landslide-free areas | 1 | 18 |
| LSI (ii) | Landslide areas | 13 | 6 |
| | Landslide-free areas | 3 | 16 |

*4.3. Landslide Risk*

Utilizing the classified LSI (i) map which appears to have better accuracy for the landslide susceptibility assessment of the area, the landslide risk analysis was designed. The risk map was developed by taking as a fact that the areas occupied by intense human presence (settlements, cultivated fields, and proximity to the road network) are ranked as moderate to high-risk zones, with continuous human presence, and are related to constant economic activities.

Based on the data inputs of hazard, exposure, vulnerability, the adopted weighting-ranking system, and the reclassification of the original values, the LR map was created (Figure 8). The LR values were grouped into five relative risk classes, which along with their percentage distribution into the study area, are demonstrated in Table 6.

**Table 6.** Percentage distribution of the five LR classes.

| Landslide Risk Classes | Very Low (%) | Low (%) | Moderate (%) | High (%) | Very High (%) |
|:---:|:---:|:---:|:---:|:---:|:---:|
| Area (percentage) | 16.60 | 26.60 | 12.10 | 39.72 | 4.98 |

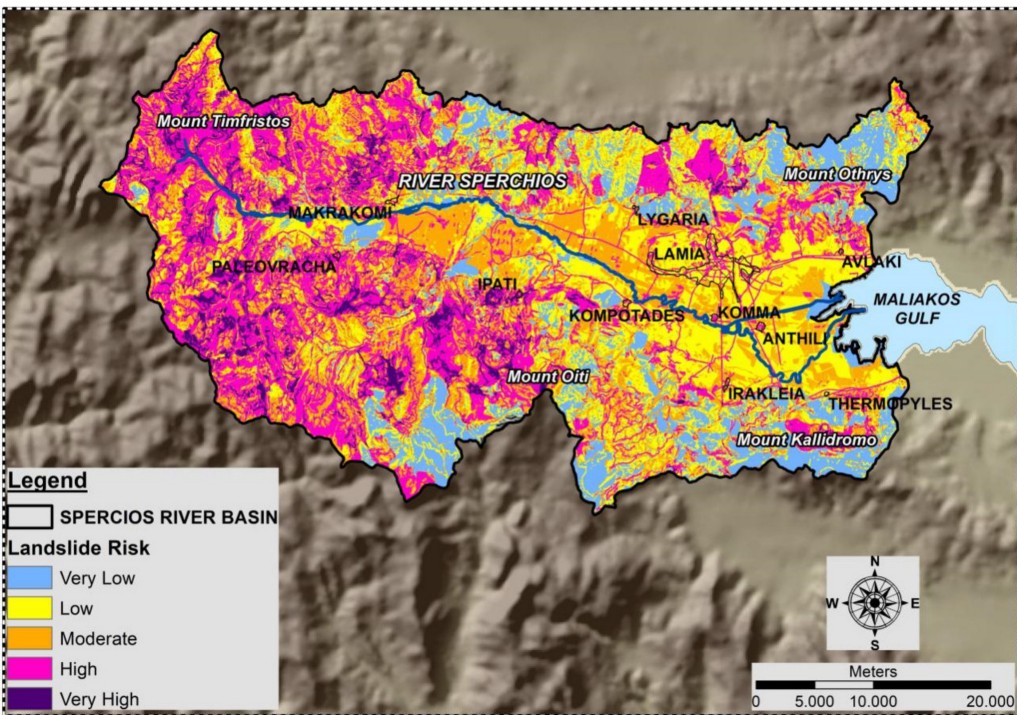

**Figure 8.** Landslide Risk map of the study area.

The statistical analysis of the LR map depicts that 44.7% of the area appears to have high and very high risk values, principally due to the increase of human activities in hilly areas nowadays. On the other hand, 43.2% of the region presents very low and low risk conditions to landslide hazard, which can be effortlessly explicated considering that the development of the main residential constructions and human activities are concentrated at the eastern coastal zone of the basin where landslides are more limited than on the southern and western mountainous regions.

### 4.4. Discussion

Landslides can trigger massive casualties and severe problems to the social and economic well-being [107,108]. The causes for landslides are numerous, complex, and every so regularly unidentified. For most morphometrical factors is challenging to be attempt to quantitatively measured them even in the field. Therefore, it is rather complicated to understand their contributions to the landslide occurrence mechanism [5,11,59,60,63]. Hence, in order to determine whether a factor affects the occurrence of landslides, several parameters were incorporated into a GIS-based analysis procedure [88,109].

The preparation of LS and LR maps is of great interest to planning organizations (at least) for introductory hazard studies, specifically when policy making is designed. Small-scale regional surveys are low-cost techniques by which larger areas can be covered in a relatively short time permitting an economical and rapid hazard assessment. Moreover, the LS map is essential for the delineation of the land use zones and the plan of future construction projects. Several methodologies have been applied for LS mapping. Eleven causative factors were considered, i.e., lithology, land use, slope angle, slope aspect, rainfall distribution, slope curvature, relative relief, proximity to faults, distance to roads, distance to streams, and soil depth. Their selection was based on the inventory map and several other supplementary data (aerial photos, satellite images, and fieldwork) [5,11,12,14,43].

Very high-resolution EO data and techniques along with Geographical Information System analysis provide powerful tools for the extraction of detailed geospatial layers, necessary for the landslide susceptibility assessment. In recent decades, many efforts have been made for integrated

and synergistic use of these innovative technologies in order to achieve more accurate timely and cost-effective results, for large geographic areas [11,28,110–114].

From the two methods applied the simple landslide index provided more accurate results. The most hazardous to landslide occurrence regions are mainly concentrated in the southwest and west area of the basin, where the most significant factors such as geology (flysch), slope, rainfall distribution, (and much less land use, relative relief, curvature, and proximity to faults) are implicated in the major landslide events [11,19,61,63,65,115]. In addition, large historically observed landslides are clearly marked in the areas of high and extremely high susceptible zones. These results show that the predicted susceptibility levels are found to be in good agreement with the past landslides. Future research efforts mandate the examination of the seismic factor influence to landslide phenomena [17,87,116], since the study area belongs to a moderate-to-high seismic risk region of Greece (according to the Seismic Risk Map created by the Greek Earthquake Planning and Protection Organization) [117]. In the present study, this venture was limited by the lack of accurate data.

## 5. Conclusions

In this study, an attempt was made to create LS and LR maps by matching AHP and WLC methodologies and evaluate their results. The semi-quantitative method (AHP–WLC) is applicable and moreover accurate for LS mapping because of the pair-wise relative comparisons of the factors without discrepancies in the decision process. Moreover, the rating values reached in this study may be used in areas of similar geological, geomorphological, and hydro-climatic conditions.

The Sperchios River basin was the object of an investigation aiming to evaluate the risk of landslide disaster by using GIS and RS technology. It is confirmed that the integration of GIS techniques and earth observation (Sentinel-2) data is decisive in productively supporting researches regarding LS and LR of local and regional areas. The land cover map creation and the comparison of Aster-derived DEM showed suitable results, confirming the quality of the subsequent analysis, as LC and many of the factors produced from the DEM, influence LS and LR.

The landslide-prone areas, delineated by the LS and LR maps, represent an essential basis for the assessment of landslide hazard and risk over the study area. Therefore, the produced maps can be particularly useful to decision-makers for choosing suitable locations for future planning in large-scale regions and similarly in disaster management planning to prepare rescue routes, service centers, and shelters.

**Author Contributions:** Conceptualization, Emmanouil Psomiadis and Nikos Charizopoulos; methodology, Emmanouil Psomiadis, Nikos Charizopoulos, Nikolaos Efthimiou., and Konstantinos X. Soulis; software, Emmanouil Psomiadis, Nikos Charizopoulos, and Ioannis Charalampopoulos; data analysis, Emmanouil Psomiadis, Ioannis Charalampopoulos, Nikos Charizopoulos, Konstantinos X. Soulis, and Nikolaos Efthimiou; resources, Emmanouil Psomiadis, Konstantinos X. Soulis, and Nikos Charizopoulos; writing—original draft preparation, Emmanouil Psomiadis, Nikos Charizopoulos, Nikolaos Efthimiou, Konstantinos X. Soulis, and Ioannis Charalampopoulos; supervision, Emmanouil Psomiadis; and field work, Emmanouil Psomiadis and Nikos Charizopoulos. All authors have read and agreed to the published version of the manuscript.

**Funding:** This research received no external funding.

**Acknowledgments:** The authors acknowledge USGS–NASA for the availability of Terra/Aster satellite data.

**Conflicts of Interest:** The authors declare no conflict of interests.

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
