# Peer review of "Earth Observation and GIS-Based Analysis for Landslide Susceptibility and Risk Assessment"

_ijgi, doi:10.3390/ijgi9090552_

Round 1

Reviewer 1 Report

It seems the authors addressed all the comments provided earlier; therefore I have no futher corrections or comments.

Reviewer 2 Report

Thank you for taking my comments into account. The manuscript is improved and I am now content.

This manuscript is a resubmission of an earlier submission. The following is a list of the peer review reports and author responses from that submission.

Round 1

Reviewer 1 Report

Reviewers feedback

The proposed manuscript is aligned with the aims and scope of this journal. It is clear, easily readable and fairly designed. It presented a GIS-based analysis of remote sensing data to assess landslide susceptibility. However, in order to send it to publish there are few improvements needed.

General comments

The manuscript is well written but few instances it failed to comply. Therefore, a professional proofreading could be a pragmatic solution. Also, some parts need to be reorganisation.

The sub-section validation should be placed under materials and method. The validation results may be presented under results and discussions with a sub-section name “accuracy assessment”.   

Conclusion is lengthy and ambiguous. I suggest rewriting this section with more clarity by synthesising main outcomes of this study.

 Specific comments

The title of this manuscript sounds a bit clunky. Could you please make it a bit transparent?

L27-29: rewrite with clarification please.

L30: GIS multicriteria analysis?

L34: a mass of rock or land mass? Confusing!

L42-43: “manifestation is of great significance”, please clarify?

L351:359: Please rewrite with clarification.

Reviewer 2 Report

Title: GIS-based multicriteria analysis and remote sensing data for landslide susceptibility and risk assessment. The case of Sperchios River basin (Central Greece)

This manuscript presents a method to assess landslide susceptibility in a given area, such as the Sperchios River basin. This technique considers 11 parameters and the Analytic Hierarchy Process (AHP) for a Weighted Linear Combination (WLC) approach.

I carefully read this paper, although the reviewer fully appreciate the time and effort authors have put into this paper, this manuscript needs minor revisions before it can be accepted. The following comments may help to improve the work of the authors.

Specific comments:

  1. The abstract needs to be revised. The aim of study must be clarified in Abstract.
  2. The novelty of this study should be better clarified in the section Introduction. The landslides should be classified as: coherent, disrupted and flow or lateral spreads.
  3. There is a need to review previous research on the paper subject. The authors are recommended to cite and discuss all these papers (for the analysis of the seismic and geologic factors):

Serey, A.; Piñero-Feliciangeli, L.; Sepúlveda, S.A.; Iveda, F.; Poblete, D.; Petley, I.; Murphy, W. Landslides induced by the 2010 Chile megathrust earthquake: A comprehensive inventory and correlations with geological and seismic factors. Landslides 2019, 16, 1153.

Chunga, K.; Livio, F.A.; Martillo, C.; Lara-Saavedra, H.; Ferrario, M.F.; Zevallos, I.; Michetti, A.M. Landslides Triggered by the 2016 Mw 7.8 Pedernales, Ecuador Earthquake: Correlations with ESI-07 Intensity, Lithology, Slope and PGA-h. Geosciences 2019, 9, 371.

Wartman, J.; Dunham, L.; Tiwari, B.; Pradel, D. Landslides in Eastern Honshu induced by the 2011 off the Pacific Coast of Tohoku earthquake. Bull. Seismol. Soc. Am. 2013, 103, 1503–1521.

Ferrario, M. F. (2019). Landslides triggered by multiple earthquakes: insights from the 2018 Lombok (Indonesia) events. Natural Hazards. doi:10.1007/s11069-019-03718-w

Livio, F., & Ferrario, M. F. (2020). Assessment of attenuation regressions for earthquake-triggered landslides in the Italian Apennines: insights from recent and historical events. Landslides. doi:10.1007/s10346-020-01464-w 

  1. The “geological” fault proximity parameter should be better analyzed, active faults or not?. To consider the seismic factor of PGA-h, macrosismic intensity (historical earthquakes) and Rrup or the closest distance to the rupture plane (ie, geological fault or subduction plane). Acceleration values (an example of, 0.30g) can be associated with a local geological fault with magnitude moderate earthquake, or a distant earthquake associated with the Hellenic subduction trench (Greece).
  2. There are many figures in the paper but the discussion and content of the paper is not enough. Also, the quality and numbered sequences of some figures could be better.

The methodology section is completed and clears enough and will help many questions in the readers mind.

Remove terms “We” in all sentences. Use passive sentences.

Reviewer 3 Report

This is a sound piece of work done well, but for me it lacks originality. In terms of the datasets used and the methods employed, this has all been done before. Indeed we used these approaches in the BGS back in the 1990s on a major study of landslides in four countries. Further evidence for this lack of original content comes from some of the key references, which are from the 1970s and 1980s. It could be published as a nice case study but I don't really feel that it takes the science anywhere new, which is a shame, because it is nicely done and well presented.

In addition, the English requires a bit of work in places. It is far better than my Greek, of course, and it is generally understandable, but a review by a native English speaker would make it just a bit more readable.

Round 2

Reviewer 1 Report

The manuscript looks sound to me. I have no further suggestions.

Reviewer 3 Report

You essentially reject both of my points. I stand by them. Therefore, we will have to agree to disagree - you have not given me any reason to change my mind!